# SUMAMOS EXCELENCIA^®^ Project: Results of the Implementation of Best Practice in a Spanish National Health System (NHS)

**DOI:** 10.3390/healthcare9040374

**Published:** 2021-03-28

**Authors:** María-Lara Martínez-Gimeno, Nélida Fernández-Martínez, Gema Escobar-Aguilar, María-Teresa Moreno-Casbas, Pedro-Ruyman Brito-Brito, Jose-Manuel Caperos

**Affiliations:** 1San Juan de Dios Foundation, San Rafael-Nebrija Health Sciences Center, Nebrija University, 28036 Madrid, Spain; mmartinezgi@nebrija.es; 2SALBIS Research Group, Faculty of Health Sciences, University of Leon, 24401 Ponferrada, Spain; 3Department of Biomedical Sciences, Faculty of Veterinary Medicine, University of León, 24071 Leon, Spain; nelida.fernandez@unileon.es; 4Nursing and Healthcare Research Unit (Investen-isciii), Carlos III Health Institute, 28029 Madrid, Spain; mmoreno@isciii.es; 5Training and Research in Care, Primary Care Management of Tenerife, The Canary Islands Health Service, 38204 Santa Cruz de Tenerife, Spain; ruymanbrito@gmail.com; 6Department of Nursing, University of La Laguna, 38200 Santa Cruz de Tenerife, Spain; 7UNINPSI, Department of Psychology, Universidad Pontificia Comillas, 28015 Madrid, Spain; jcaperos@nebrija.es; 8Fundación San Juan de Dios, 28036 Madrid, Spain

**Keywords:** pain, accidental falls, urinary incontinence, evidence-based clinical practice, clinical audit

## Abstract

The use of certain strategies for the implementation of a specific recommendation yields better results in clinical practice. The aim of this study was to assess the effectiveness of an evidence-based model using clinical audits (GRIP model), for the implementation of recommendations in pain and urinary incontinence management as well as fall prevention, in the Spanish National Health System during the period 2015–2018. A quasi-experimental study has been conducted. The subjects were patients treated in hospitals, primary care units and nursing home centers. There were measures related to pain, fall prevention and urinary incontinence. Measurements were taken at baseline and at months 3, 6, 9, and 12. The sample consisted of 22,114 patients. The frequency of pain assessment increased from 59.9% in the first cycle to a mean of 71.6% in the last cycle, assessments of risk of falling increased from 56.8% to 87.8% in the last cycle; and finally, the frequency of assessments of urinary incontinence increased from a 43.4% in the first cycles to a mean of 62.2% in the last cycles. The implementation of specific evidence-based recommendations on pain, fall prevention, and urinary incontinence using a model based on clinical audits improved the frequency of assessments and their documentation.

## 1. Introduction

For more than four decades, studies have reported the existence of a large gap between research and clinical practice. This means that research results are not contributing to the improvement of care, and there is a high level of variability which complicates decision making and implementing the “best care” available. This happens despite the greater awareness of and accessibility to evidence [1,2].

This lack of consensus increases the risk of making mistakes, the inappropriate use of resources, and the continuation of practices that are ineffective or even potentially harmful [1,3]. However, the use of research is a complex process, in which the characteristics of the healthcare professionals, the organizations (context) and the innovation itself are related [3,4,5].

In addition, it seems increasingly evident that the processes of dissemination and implementation of scientific knowledge are not passive, rather they require active strategies to ensure that evidence is properly understood and effectively adopted, implemented, and maintained in clinical practice settings [6].

Improving the understanding of the impact of contextual factors, by finding ways to integrate them into strategy design, is a key element of evidence implementation strategies. A recommended approach is to conduct a situational analysis, identify barriers before starting an implementation project, and develop specific interventions to decrease the impact of these barriers [7]. A Cochrane systematic review stated that future implementation studies should address how barriers are identified and how to overcome them, which should be part of any strategy [8].

There are studies that base their implementation on simple interventions, such as distributing printed materials or holding targeted educational meetings, but their effectiveness is very limited. The use of multi-component strategies may be effective for improving nurses’ knowledge and behaviors, but their effect on patients has rarely been measured [9]. In a study by Powell et al. (2012) they found 68 implementation strategies, including encouraging stakeholder participation, training professionals to perform the intervention, conducting audits, providing feedback, and creating standards [10].

Some systematic reviews have assessed the effect of introducing Clinical Practice Guidelines (CPG) in healthcare, but great variability has been found in the results obtained, depending on the indicators and guidelines implemented; which may be related to the complexity of the implementation process [11,12,13,14]. Using strategies and resources to implement specific recommendations from a Clinical Practice Guideline provides good results in clinical practice, as it does not require major organizational changes at the institutional level [8]. Conducting clinical audits and providing feedback, may be effective strategies for improving the quality of healthcare services and for the implementation of best practice, as they help some professionals demonstrate their willingness to change their behaviors [8,15]. Clinical audits are useful as a strategy within a continuous quality improvement process. The strategy revolves around measuring a clinical outcome or a process against well-defined standards, established using the principles of evidence-based practices [16,17]. Audit and feedback is widely used as an strategy to improve professional practice and to attenuate practice variation among professionals [18,19].

Likewise, studies on interventions for the implementation of nursing practice quality circles significantly improved levels of care overload and the use of innovation in the workplace [20].

This study is based on the GRIP [21] model, which is a quality improvement process including a previous basal clinical audit (Figure 1).

The GRIP model was developed by the Joanna Briggs Institute for Evidence-based Health Care. It is a model designed to establish interprofessional processes within teams, to examine the barriers (obstacles) that hinder the use of evidence in promoting best practice; and to help develop actions to be implemented to overcome these barriers [21]. Smaller-scale studies have used audit and feedback based on the GRIP model, and the results have been favorable [22,23,24].

In this context, proposals for change that require the involvement of the team, the detection of difficulties in achieving change, the selection of actions to overcome barriers, and the definition and monitoring of indicators to measure success, are necessary elements for changing practice and implementing evidence [25].

In 2012, the Nursing & Healthcare Research Unit and the Spanish Collaborating Centre of the Joanna Briggs Institute created a national network of centers targeted at implementing Clinical Practice Guidelines (CPG) in healthcare. This project has shown that implementation of evidence-based recommendations from CPG improves processes in clinical practice and has positive effects on patient outcomes [26,27,28].

A project was started, as a way of continuing the research program being pursued by both the above-mentioned centers, in which nursing staff and other health professionals could implement specific evidence-based recommendations, instead of implementing full CPGs; this project is Sumamos Excelencia® [29]. 

Pain, falls and urinary incontinence (UI) are health problems that are highly prevalent in the general population, especially in individuals over 65 years of age, and there is strong evidence to support their requirement for nursing care. However, these health problems are scarcely assessed, managed, and monitored in current clinical practice [30,31]

According to International Association for the Study of Pain, pain is defined as “An unpleasant sensory and emotional experience associated with, or resembling that associated with, actual or potential tissue damage”. Pain has a profound impact on the quality of life and may have physical, psychological and social consequences. It has globally been estimated that one in five adults suffer from pain. Moreover, this prevalence increases in the institutionalized population subjected to painful procedures [32]. According to the US Joint Commission’s standards of care, all patients should be evaluated, treated and monitored of their pain level. Even so, in current clinical practice assessment of pain is infrequent, as is its management and monitoring [31,33].

Urinary incontinence (UI) is the involuntary loss of a person’s bladder control at an inappropriate time and place, and it is known to cause in the person hygienic, social and psychological difficulties [34]. Urinary incontinence is one of the biggest problems affecting older people and is of great significance in public health [35]. According to the majority of studies, the estimated prevalence of urinary incontinence ranges from 23% to 44%. According to Moehrer [36], approximately 20% of women aged 25–64 years experience some UI-related symptom.

Depending on its form of presentation, UI can be urge incontinence, when there is an urgent sensation of bladder emptying that is difficult to postpone; stress incontinence, when the loss coincides with an increase in pressure in the intra-abdominal cavity; and mixed incontinence, in which there is a combination of the two [37].

On the other hand, falls are one of the principal cause of preventable adverse events in health institutions, and are the second leading cause of death due to accidental or unintentional injuries worldwide [38]. It is established that each year one third of the population over the age of 65 years, experience a fall, a proportion that rises among institutionalized patients [39], and most fatal falls are suffered by people over 65. Approximately, 28–35% of elderlies aged 65, and older, fall every year, and that number increases to 32–42% for people over 70. Organizations should evaluate their patients’ risk of falling, in the context of the attended population, care should be offered and they should take measures to reduce the risk of falls and fall-induced injuries [33].

The purpose of this study is to assess a model for the implementation of specific evidence-based recommendations, as measured using clinical audits, that will enable a better assessment of patients and a reduction of the practice variation in decision making. We also aim to determine the baseline degree of compliance with recommendations in the participating units and nursing home centers; to analyze the initial and final levels of compliance with the best practice criteria; to analyze the implementation barriers detected in the participating units and nursing home centers, and the actions to be taken to overcome them.

### Hypothesis

The implementation of evidence-based recommendations using the GRIP model improves the assessment and management of pain, urinary incontinence, and fall prevention in patients [22,23,24].

## 2. Materials and Methods

### 2.1. Design

This is a quasi-experimental multi-centered study. The intervention consisted in the application of recommendations in the chosen issue by each unit, with outcome measurements taken at baseline and, subsequently, 3, 6, 9, and 12 months after the baseline measurement was taken.

### 2.2. Setting and Participants

The study was conducted in healthcare units and nursing home centers of the Spanish National Health System (SNHS) between 2015 and 2018. Patients seen in SNHS units (primary care, hospitals) and nursing home centers. In Spain, the nursing home centers are institutions, where care is provided in form of personal assistance in daily living, as well as nursing and medical care, for older people who live there for an undefined period of time.

#### 2.2.1. Inclusion Criteria for the Units

Units that committed themselves to implementing recommendations regarding one of the following issues: pain, falls, or urinary incontinence. For the purposes of this study, “units” were considered to be those services (primary care), centers (hospitals), or institutions (nursing homes) providing healthcare services to patients who share similar characteristics.

#### 2.2.2. Inclusion Criteria for Patients

Patients seen in the participating units. Pain: individuals admitted to hospitals, primary care, or nursing home institutions who could potentially suffer from some type of pain. Falls: individuals admitted to hospitals, primary care, or nursing home institutions, age of 65 or over, presenting one or more risk factors for falls according to the assessment criteria established in the risk measurement instruments used. Urinary incontinence: individuals residing in the community or institutionalized individuals who are susceptible to urinary incontinence.

#### 2.2.3. Recruitment of Study Subjects and Sample Size

Units: intentional non-probabilistic sampling. A participation process was launched and published through the channels provided by the Nursing and Healthcare Research Unit (Investen-isciii) and the Spanish Center for Evidence-Based Nursing and Healthcare: which is a JBI Centre of Excellence. We selected the units that met the inclusion criteria and agreed with the project requirements. It was estimated that 100 units of each group of recommendations should be included, which would account a total of 300 units.

All patients who met the inclusion criteria when data were collected from their unit were included. All data were collected during the last five working days of each evaluation quarter, to homogenize the collection periods. Data collected from different types of units were as follows: Primary care units: patients seen in consultations during the last five working days of each assessment trimester. Hospital units: patients discharged in the last five working days of each assessment trimester. Nursing home centers: patients admitted in the last five working days of each assessment trimester.

### 2.3. Intervention

The intervention consisted of the implementation of specific evidence-based recommendations regarding the assessment and management of pain, the detection and management of urinary incontinence, and the fall prevention through assessment and management of the risk of falling (Appendix A). This was done using a multi-component strategy that included: training, facilitators, auditing and feedback.

Phase 0: Recruitment. In this phase, the units were recruited through a public call for participation. Phase 1: Preparation. This phase was guided by external and internal facilitators. The external facilitators were the project researchers who provided training and methodological and logistical support to the participating units. The internal facilitators (researchers responsible) were nursing professionals from the participating units who were identified as project leaders within each unit. They received, along with the other professionals from the participating unit, an intensive online training program addressing the following: knowledge of evidence-based practice, evidence-based recommendations on the topic to be implemented, process and outcome indicators, and management of clinical audits. The training program lasted two months and ended with a knowledge assessment test. Internal facilitators were trained in the use of the online data collection platform to ensure consistent data collection. Phase 2: Implementation. The implementation phase was based on reports from the participating units about the baseline assessment of compliance to evidence-based recommendations specific to the health problem to be addressed. An analysis of the situation: identification of barriers to achieving compliance, identification of actions to address these barriers, and resources associated with each action of improvement. Incorporation of recommendations into clinical practice in as many as permitted from all the barriers detected. An online data collection platform was created for the project, which generated reports with the data on compliance, as well as recommendations, an action plan with the barriers identified, the actions to be implemented, the responsible parties, the necessary resources, and a date. Phase 3: Follow-up. After a baseline assessment, the process was monitored with measurements at month 3, 6, 9, and 12 while continuing to identify barriers and establish actions for improvement.

### 2.4. Measurements

The data related to the indicators regarding the assessment, detection and management of pain, fall prevention and urinary incontinence was collected. All indicators were selected based on CPGs and relevant scientific evidence [40,41]. In Spain there were no updated guidelines in the context of nursing care. The content of these guides has been translated, adapted and used previously in projects for the implementation of evidence in the Spanish healthcare context [27,42]. The indicators and variables used can be consulted in the study protocol [29] (Appendix A). The primary outcomes were the assessment in pain, fall risk and urinary incontinence, the second outcomes were the implementation of care plan and education.

### 2.5. Data collection

Data collection took place over a period of 15 months (April 2016–September 2017). Due to a delay in the beginning of the project, measurements corresponding to months April and September were considered baseline (units training period), as neither unit had performed the implementation of any of the recommendations by that time. The measurements from April 2016 (baseline), September 2016 (baseline), December 2016 (3 month), March 2017 (6 month), June 2017 (9 month), and September 2017 (12 month) were considered to be a complete measurement cycle (September–2017 cycle was optional). At each point in the cycle, all indicators were measured according to the issue to be implemented: pain, falls or UI. Each issue included process indicators (assessment, management, care plan, education) and outcome indicators.

All the units that completed the monitoring for at least nine months were considered for analysis. An initial audit of all the indicators was conducted to determine the degree of compliance to the evidence-based recommendations included in the project. Subsequently, a follow-up was conducted every three months (at month 3, 6, 9, and 12) to reassess compliance with all the indicators measured in the initial audit (April-September). Patient and indicator data were collected by the responsible researchers (internal facilitators) for each unit. They were collected from the medical and nursing records and entered into the platform created for the project.

### 2.6. Data Analysis

Initially, we calculated the percentages (with a 95% confidence interval) of compliance for each item (assessment, care plan, patient education, comprehensive assessment and restraining measures) and for each health problem assessed (pain, fall prevention, and urinary incontinence). Subsequently, due the dichotomic nature of the outcomes, we analyzed the changes between the cycles using a generalized logistic mixed methods model including units nested within hospitals as a random factor and patients’ sex and age as covariates. Finally, in order to assess the impact of the intervention, we calculated the percentages of compliance in each unit and the estimated average change observed from baseline cycles (cycles 0 and 1) to post-implementation cycles (cycles 4 and 5) using a mixed methods model to account for dependency between units within the same hospital.

Analyses were performed using the Lme4 packages (Bates et al., 2015)[43] and LmerTest (Alexandra Kuznetsova et al. 2020) [44] in the R programming language (Pinheiro, Bates, DebRoy, Sarkar, and R Core Team 2015)[45]. The statistical significance threshold for the results was set at *p* < 0.05, with a 95% confidence interval for all cases.

## 3. Results

### 3.1. Description of the Sample

A total of 288 units responded to the recruitment process, of which a total of 135 finally participated. Figure 2 is a flow chart of the recruitment process.

A total of 22,114 patients were analyzed, whose distribution and characteristics are reflected in Table 1.

### 3.2. Primary Outcome: Implementation of a Model for the Assessment of Pain, Risk of Falling and Urinary Incontinence

In the case of pain assessment, the final sample consisted of 10,192 patients. We found that the frequency of pain assessment increased from 59.9% in the first cycle to a mean of 71.6% in the last cycle (*Z* = 15.88; *p* < 0.001) (Table 2 and Table 3). We found no differences in pain assessment in relation to patients’ sex (*Z* = 1.00; *p* = 0.318) or age (*Z* = 0.63; *p* = 0.531) (Table 2).

In the case of risk of falling, the final sample was 7782 patients. We found an increase in the frequency of assessments of risk of falling across study cycles, from 56.8% in the first cycle to a mean of 87.8% in the last cycle (*Z* = 20.49; *p* < 0.001) (Table 2 and Table 3).

We also found a positive effect of the age factor, since older patients were more frequently assessed than younger patients (*Z* = 6.58; *p* < 0.001). We did not find sex differences in fall risk assessment (*Z* = 0.40; *p* = 0.692) (Table 2).

Finally, in the case of urinary incontinence, the final sample was 2492. We found an increase on the frequency of urinary incontinence assessments across study cycles from a 43.4% in the first cycles to a mean of 62.2% in the last cycles (*Z* = 10.87; *p* < 0.001) (Table 2 and Table 3). We also found a positive effect of the age on the assessments: older patients were more frequently assessed than younger patients (*Z* = 7.16; *p* < 0.001). We did not find sex differences in urinary incontinence assessments (*Z* = −0.10; *p* = 0.917) (Table 2).

### 3.3. Secondary Outcome: Efficacy of the Implementation Model in the Provision of Care Plans and Patient Education

Regarding pain secondary outcomes, we also found increases in care plan implementation (*Z* = 16.53; *p* < 0.001), frequency of patient education (*Z* = 25.30; *p* < 0.001), and comprehensive pain assessment (*Z* = 15.99; *p* < 0.001) across study cycles (Table 3). In the case of risk of falling, as well as we found an increase in the implementation of the care plan (*Z* = 11.52; *p* < 0.001) and restraining measures (*Z* = 4.407; *p* < 0.001) across study cycles. We found a positive effect of age regarding restraining measures (*Z* = 16.41; *p* < 0.001) and a marginal non-significant trend on the implementation of the care plans (*Z* = 1.94; *p* = 0.052). Finally, in the case of urinary incontinence, we found as well an increase in the implementation of care plans (*Z* = 4.72; *p* < 0.001) and in the frequency of patient education (*Z* = 6.44; *p* < 0.001) across study cycles.

### 3.4. Impact of the Intervention

In order to assess the impact of the intervention, we aggregated data from baseline cycles (0 and 1) and post-intervention cycles (4 and 5) and calculated the mean change and Cohen’s effect size (*d*). We found medium to large changes in all evaluated outcomes except in the implementation of restraining measures, with the largest changes occurring in the field of pain. The intervention caused increases of up to 30–40% (e.g., in the development of patient education), and over 15–20% in patient evaluation in all fields.

In the case of pain, we found a mean increase of 18.4% in assessments (*d* = 0.68), 24.1% (*d* = 0.67) in the implementation of care plans, 40.3% (*d* = 1.13) in educational information, and 21.8% (*d* = 0.61) in comprehensive assessments of pain. Regarding risk of falling, we found a mean increase of 15.6% (*d* = 0.55) in assessments, 16.0% (*d* = 0.48) in the implementation of care plans, and 4.6% (*d* = 0.19) in restraining measures. Finally, related to urinary incontinence, we found a mean increase of 20.3% on assessments (*d* = 0.58), a 21.4% (*d* = 0.33) in the implementation of care plans, and 35.6% (*d* = 0.97) in educational information.

### 3.5. Barriers Detected during Measurement Cycles and Planned Actions

Forty-six units (34%) reported barriers, with a total of 218 barriers and 218 actions. The most reported barrier was the lack of recording tools for the clinical records (30.3%, *n* = 66), followed by the lack of action procedures (25.2%, *n* = 55) and the incompletion of records (19.7%, *n* = 43). Regarding the reported actions, training of professionals (52.3%, *n* = 114) was the most implemented, followed by the creation of computer tools for the registers (39%, *n* = 85). The frequencies of the barriers and actions implemented by the units are shown in Table 4.

## 4. Discussion

SUMAMOS EXCELENCIA^®^ is a Spanish national project that aims to create a network for the translation of evidence into practice through specific recommendations from CPGs.

In this project, a model for implementing evidence-based recommendations (Getting Research into Practice-GRIP) has been evaluated. To this end, we used a multi-component strategy that integrates educational elements (training of professionals), the figure of facilitators (both internal and external), the performance of audits and the provision of feedback. This model has previously been used in other studies with a smaller number of units [23,24,46,47]. In this first cohort, a total of 135 units participated and a total of 22,114 patients were analyzed. This has meant the involvement in the project of more than 500 nursing professionals from all over the SNHS.

### 4.1. Baseline Situation of the Degree of Compliance with the Recommendations in the Participating Units and Final Levels of Compliance to the Best Practice Criteria

At baseline, the participating units complied with recommendations in the 59.93%, 56.8%, and 43.4% of the patients for pain assessment, risk of falling, and detection of urinary incontinence, respectively. With regard to pain assessment, the study by Stevens et al. reported assessment percentages of around 67–69% [48]. Perhaps this difference with respect to our study may be explained by the pediatric setting of their research, where there is greater awareness of pain assessment. The study by Daniels reported baseline frequencies of 56%, and another similar study, conducted in the context of cancer pain with the same implementation model, reported a baseline assessment rate below 50% [46]. This indicates that, despite the Joint Commission’s consideration of pain assessment as the fifth vital sign, there is still an under-assessment of pain in clinical practice [33].

In regard to the risk of falling, a study conducted with the same implementation model, reported baseline frequencies in the assessment of the risk of falling between 90–93%, depending on the type of unit [24]. Similarly, a study by Johnson et al. indicated a baseline assessment of 95% [49]. In SUMAMOS EXCELENCIA^®^ we started from lower baseline levels (56.8%). This may be due to the fact that the project includes different types of units (medical, surgical, medical-surgical) and different types of centers (hospitals, primary care centers, and nursing home centers), while the rest of the studies focused on more homogeneous centers, which may decrease variability. Lastly, regarding to the assessment of urinary incontinence, a study using the same implementation model reported baseline data of 100% compliance in assessing urinary incontinence [23]. Perhaps because it was carried out in a medium- to long-term hospital for patients over 65, where the assessment criteria for urinary incontinence are more present, given that it is a highly prevalent problem in individuals over 65.

Concerning the final levels of compliance to the evidence-based recommendations and taking into account the studies mentioned previously, in SUMAMOS EXCELENCIA^®^ the assessment of pain has increased by 11.7%. Similar studies report increases ranging from 13% in Daniels et al., 25% in Stevens et al., 33% in Ang & Chow and to 43% in Dulko et al. [22,46,48,50]. This variability in final compliance may be conditioned by the baseline data used, which were lower in the study reporting the greatest increase [50]. The results of the study by Daniels are the closest to our results, with similar baseline data. The increase in compliance was between 13 and 43%. This 30% range coincides, although with a more optimistic interval, with that reported by Lau et al., who pointed out that only a third of the evidence containing recommendations was routinely followed, with compliance to recommendations ranging from 20 to 80% [51].

Regarding the assessment of the risk of falling, which was increased by 30.7% in our study, the rest of the studies showed a smaller increase, between 3% and 10% in the case of Szymaniak [24], and even a decrease of 1% in the case of Johnson et al. [49].

Finally, concerning the detection of urinary incontinence, the increase in compliance in our study was 18.8%. The only study available which was similar to our intervention reported no increase in compliance, since it started from a baseline value of 100% [23].

### 4.2. Assessment Indicators, Barriers and Actions in the Implementation Process

This project is based on a model that integrates simple and specific indicators. The homogenization of indicators implies a decrease in inter-professional variability; moreover, benchmarking will allow professionals to measure results and evaluate the quality of the care provided [52]. These indicators were used in other implementation studies based on the GRIP model [22,23,24].

On the other hand, previous studies recommend ”mapping the territory” before starting an implementation project and developing specific actions to identify barriers, thus reducing their impact. In the project SUMAMOS EXCELENCIA^®^, barriers were identified, among which the lack of specific registration tools in clinical records and the lack of training in professionals stand out. As a result, the majority of the actions undertaken were focused on these aspects. This is in line with other studies that highlight professional training as one of the barriers detected and which design educational interventions to implement the evidence available [13,22,23,24]. Identifying barriers and taking action enables improvements to be made in the current knowledge of the causal mechanisms behind the success or failure of an intervention [1,4]. Indeed, a 2010 Cochrane systematic review recommends describing explicitly how to identify these barriers and propose actions to overcome them [53].

### 4.3. Facilitators in Implementation Projects

The figure of the facilitator is a contextual element included in the PARiHS model [54]. This figure has been used in other studies, reporting that facilitation based on the facilitator’s role and the relationship established with the individuals, may affect the changes [22,23,24,48].

This project has included both internal facilitators (well-trained staff of the participating units), who prepare, guide, and support the members of their team during the implementation process, and external facilitators professionals from the research team with experience in projects implementing evidence into clinical practice [26,27]. External facilitators were responsible for training the internal facilitators and monitoring their performance throughout the project.

### 4.4. Clinical Audits and Feedback as Implementation Strategies

The GRIP model used in this project is based on a process of improvement in response to a previous baseline clinical audit. Interventions using educational activities and audits are the most widely used, although the effectiveness of both remains unclear [55]. Audits and feedback have been shown to change behaviors in physicians, and it is recommended that their effectiveness in the field of nursing should be studied [7,56]. Within the field of nursing, there are studies that have used auditing and feedback based on the GRIP model on a smaller scale, with favorable results [22,23,24]. For the audit and feedback process to be effective, it is important that the professionals involved understand the purpose of the audit [19,55]. The project SUMAMOS EXCELENCIA^®^ provides a pre-training stage where internal facilitators and unit professionals receive online training in clinical auditing and evidence-based practice. This enables professionals to understand the purpose of the audit and how to manage it. Each unit adapted the project to its local circumstances, allowing professionals to perceive the opportunity for change provided by the project and thus adjust it to the needs of each unit.

### 4.5. Limitations

The conclusions of this study may be limited by the biases inherent in a non-controlled and non-randomized design. The data should therefore be interpreted with this in mind. The necessary sample size was estimated to be 100 units for each health problem; in the case of pain, 63% of the sample size was reached, with 55% for falls and 17% for urinary incontinence. It is true that the calculation was an estimate without reference data, which is a limitation of our study. The low response rate in the area of urinary incontinence may be related to two elements: on the one hand, the low priority given to the treatment of urinary incontinence, considering it as part of ageing and difficult to manage; on the other hand, the fact that the participating units were asked to choose only one of the three areas (pain, urinary incontinence or falls) which may have influenced the choice of the other areas in detriment of incontinence [57,58].

There were contextual elements that may have influenced the improvement described, one of them being the monitoring process inherent in the audit procedure, which makes the participants aware of the process and may lead them to be affected by the Hawthorne effect. However, with a follow-up of 12 months, this effect is estimated to be minimal.

## 5. Conclusions

This study shows that the implementation of specific evidence-based recommendations on assessment pain, fall prevention, and urinary incontinence, using a model based on clinical audits, improves the frequency of assessments and their documentation.

It also demonstrates that an easy-to-apply model, based on clinical audits, contributes to improved pain assessment and management, fall prevention and detection of urinary incontinence.

In the project SUMAMOS EXCELENCIA^®^, barriers have been identified, among which the lack of specific registration tools in clinical records and the lack of training in professionals stand out. As a result, the majority of the actions undertaken were focused on these aspects.

For this project we used a multi-component strategy that integrates educational elements (training of professionals), the figure of facilitators (both internal and external), the performance of audits and the provision of feedback. These components are key elements of evidence implementation strategies.

## Figures and Tables

**Figure 1 healthcare-09-00374-f001:**
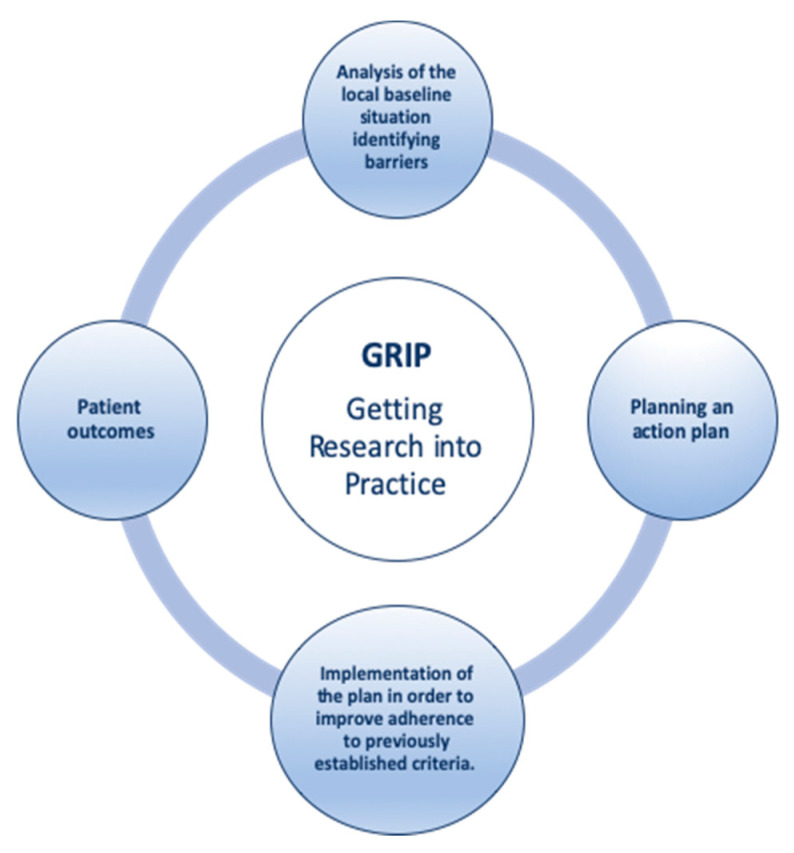
GRIP model (Getting Research Into Practice) [21].

**Figure 2 healthcare-09-00374-f002:**
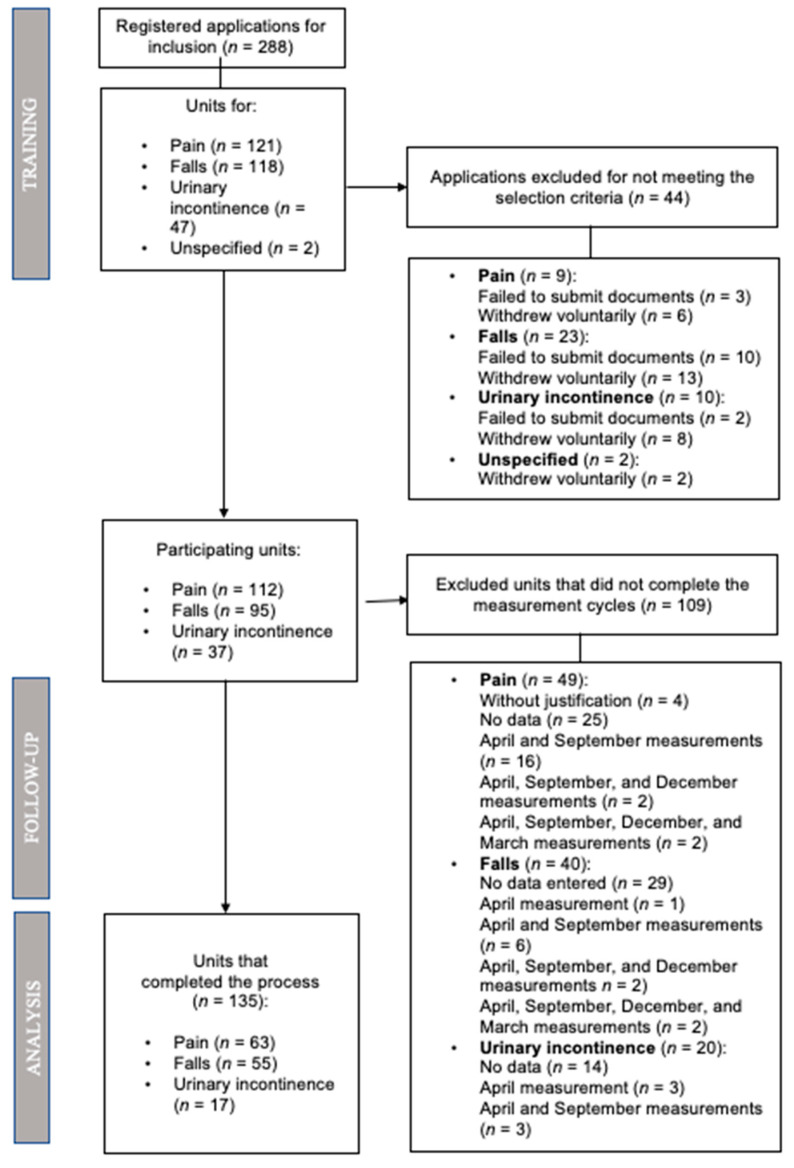
Flow chart of the unit recruitment process in Sumamos Excelencia.

**Table 1 healthcare-09-00374-t001:** Characteristics of the units and the patients.

Type of Unit	Pain		Fall Prevention		Urinary Incontinence		Total
Medical	19		30		9		58
Surgical	14		5		1		20
Medical-surgical	13		8		0		21
Critical care	8		0		0		8
Primary care/outpatient care	2		7		3		12
Maternal and childcare	2		0		1		3
Other (emergency care and delivery care)	4		1		0		5
Nursing home	1		4		3		8
Total	63		55		17		135
Complete cycle 9 month	9		6		4		9
Complete cycle 12 month	54		49		13		116
Mean age	55.2 (SD: 23.13)		75.93 (SD: 14.13)		69.12 (SD: 17.55)		66.75 (SD: 18.27)
**Sex**							
Male	5661	49.5%	4462	55.0%	1074	41.8%	11,197
Female	5595	48.9%	3617	44.6%	1481	57.7%	10,693
Missing data	184	1.6%	27	0.3%	13	0.5%	224
**Patients per cycle**	*n*		*n*		*n*		*N*
Cycle 0 (April-baseline)	2153		1608		481		4242
Cycle 1 (September-baseline)	1851		1372		447		3670
Cycle 2 (December-3 months)	2050		1565		448		4063
Cycle 3 (March-6 months)	2040		1334		417		3791
Cycle 4 (June-9 months)	2075		1243		400		3718
Cycle 5 (September-12 months-optional)	1271		984		375		2630
TOTAL	11,440		8106		2568		22,114

**Table 2 healthcare-09-00374-t002:** Percentages (95% CI) of compliance for each item in the cycles.

Issues	Cycle					
*Pain*	0	1	2	3	4	5
Assessment	59.9 [57.8–62.1]	57.8 [55.5–60.2]	61.7 [59.6–63.9]	73.6 [71.7–75.6]	70.2 [68.2–72.2]	71.6 [69.1–74.2]
Care plan	51.1 [47.6–54.7]	55.1 [51.3–59.0]	56.7 [53.3–60.1]	69.3 [66.3–72.3]	74.0 [71.1–77.0]	73.1 [69.8–76.4]
Patient education	12.2 [9.90–14.6]	16.9 [14.0–19.8]	30.0 [26.9–33.1]	50.8 [47.6–53.9]	65.5 [62.4–68.7]	68.4 [64.9–72.0]
Comprehensive assessment	69.6 [66.3–72.9]	59.3 [55.5–63.1]	74.1 [71.1–77.1]	83.9 [81.5–86.2]	84.5 [82.1–86.9]	78.7 [75.5–81.8]
*Fall prevention*						
Risk assessment	56.8 [54.4–59.3]	56.1 [53.4–58.8]	69.8 [67.5–72.1]	78.3 [76–80.5]	76.6 [74.2–79.0]	87.8 [85.7–89.9]
Care plan	65.8 [61.9–69.7]	67.3 [63.2–71.5]	82.3 [79.5–85]	77.5 [74.3–80.6]	82.6 [79.4–85.8]	84.5 [81.4–87.6]
Restraining measures	39.4 [36.8–42.0]	35.8 [33.0–38.6]	41.2 [38.5–43.8]	41.4 [38.6–44.3]	39.5 [36.5–42.5]	33.9 [30.7–37.1]
*Urinary incontinence*						
Assessment	43.4 [38.9–47.9]	41.7 [37.1–46.3]	52.7 [48.1–57.4]	62.3 [57.6–67.1]	72.3 [67.9–76.8]	62.2 [57.2–67.2]
Care plan	68.5 [57.8–79.1]	56.5 [44.8–68.2]	63.6 [54.4–72.7]	66.7 [57.6–75.7]	86.2 [79.9–92.5]	80.9 [72.9–88.8]
Patient education	41.1 [29.8–52.4]	33.3 [22.0–44.7]	25.7 [17.4–34.1]	24.8 [16.3–33.2]	70.5 [62.1–79.0]	60.9 [50.9–70.8]

**Table 3 healthcare-09-00374-t003:** Logistic regression models of compliance as a function of the cycle and patients’ sex and age.

Issues	Pain			Fall Prevention			Urinary Incontinence		
	Adjusted OR for Evaluation	*z-*Value	*p-*Value	Adjusted OR for Evaluation	*z-*Value	*p-*Value	Adjusted OR for Evaluation	*z-*Value	*p-*Value
*Primary outcome*									
Intercept	1.33 [0.78–2.28]	1.036	0.300	0.60 [0.29–1.25]	−1.356	0.175	0.43 [0.12–1.57]	−1.271	0.204
Cycle	1.30 [1.26–1.35]	15.877	< 0.001	1.50 [1.44–1.56]	20.493	< 0.001	1.41 [1.32–1.49]	10.869	< 0.001
Sex (female)	1.05 [0.95–1.17]	0.998	0.318	1.02 [0.91–1.15]	0.396	0.692	0.99 [0.80–1.22]	−0.104	0.917
Age (years)	1.00 [1.00–1.00]	0.626	0.531	1.02 [1.01–1.02]	6.583	< 0.001	1.03 [1.02–1.04]	7.161	< 0.001
*Secondary outcome 1*									
Intercept	1.61 [0.45–5.71]	0.731	0.465	1.12 [0.31–3.95]	0.169	0.866	7.8 [0.89–68.74]	1.851	0.064
Cycle	1.85 [1.72–1.99]	16.526	< 0.001	1.57 [1.45–1.69]	11.521	< 0.001	1.50 [1.27–1.77]	4.718	< 0.001
Sex (female)	1.31 [1.06–1.62]	2.525	0.012	1.02 [0.81–1.28]	0.165	0.869	1.48 [0.85–2.56]	1.391	0.164
Age (years)	0.99 [0.98–0.99]	−4.386	< 0.001	1.01 [1.00–1.02]	1.944	0.052	0.98 [0.96–1.00]	−1.632	0.103
*Secondary outcome 2*									
Intercept	0.07 [0.02–0.21]	−4.855	< 0.001	0.00 [0.00–0.01]	−11.832	< 0.001	1.04 [0.11–9.82]	0.038	0.969
Cycle	2.82 [2.6–3.05]	25.295	< 0.001	1.09 [1.05–1.14]	4.407	< 0.001	1.74 [1.47–2.05]	6.435	< 0.001
Sex (female)	1.01 [0.83–1.22]	0.06	0.952	1.11 [0.97–1.27]	1.551	0.121	1.96 [1.08–3.56]	2.203	0.028
Age (years)	1.00 [0.99–1.00]	−1.793	0.073	1.06 [1.05–1.07]	16.415	< 0.001	0.97 [0.94–0.99]	−2.819	0.005
*Secondary outcome 3*									
Intercept	4.13 [1.14–15.06]	2.152	0.0314						
Cycle	1.85 [1.71–1.99]	15.989	< 0.001						
Sex (female)	0.77 [0.62–0.96]	−2.29	0.022						
Age (years)	0.99 [0.99–1.00]	−1.997	0.046						

Primary outcome = assessment of pain and urinary incontinence and of the risk of falling in fall prevention; Secondary outcome 1 = implementation of care plan in all areas; Secondary outcome 2 = education for pain and urinary incontinence and the implementations of restraining measures to reduce risk of falling; Secondary outcome 3 = comprehensive assessment of pain. OR: Odds Ratio.

**Table 4 healthcare-09-00374-t004:** Description of the barriers identified and actions applied.

Barriers Identified				Actions Undertaken			
Type of Barrier		No.	%	Institutional Support	Tool Development	Patient-Family Education	Training of Professionals
***Context***							
	Lack of tool for computer registration	66	30.3%	-	58	1	7
	Lack of action procedures	55	25.2%	3	21	10	21
	Type of patient	10	4.6%	-	1	1	8
	Time	2	0.9%	1	-	-	1
	Difficulty in handling tools	1	0.5%	-	-	-	1
***Individual***							
	Incomplete records	43	19.7%	-	5	-	38
	Lack of training	36	16.5%	-	-	2	34
	Lack of interest	5	2.3%	1	-	-	4
***Result***		218	%	2.3%	39.0%	6.4%	52.3%

## Data Availability

The data presented in this study are available on request from the corresponding author. The data are not publicly available due to privacy/ethical restrictions.

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
