# Peer review of "SUMAMOS EXCELENCIA® Project: Results of the Implementation of Best Practice in a Spanish National Health System (NHS)"

_healthcare, 2021, doi:10.3390/healthcare9040374_

Round 1

Reviewer 1 Report

Revision of the article:

SUMAMOS EXCELENCIA® Project: Results of the implementation of best practice in a Spanish National Health System (NHS)

This article aims to assess a model for the implementation of specific evidence-based recommendations as measured using clinical audits following the GRIP model. It is a well designed and developed article that studies the baseline degree of compliance with recommendations in the participating units and nursing home centers regarding assessment and management of pain, the detection and management of urinary incontinence and the fall prevention through assessment and management of the risk of falling. It is also analyzed the initial and final levels of compliance to best practice criteria, as well as the barriers and the actions to be taken to overcome them.

It is a well-conducted study that has important implications for practice and for improving care.

I would like to congratulate the researchers for conducting this study and at the same time I would like to propose some area for improvement prior to publication:

  • In the limitations section, it would be desirable to explain in greater depth what was the reason or the difficulty to finally have only 17 units to assess urinary incontinence.
  • At the format level, in the discussion section some references are cited according to the APA style and not the style recommended by Healthcare journal. It is recommended to review this aspect.
  • It is also recommended to review the References section since some references are not well referenced (see for example reference 51 on lines 597-598). You can check the recommendations in https://www.mdpi.com/journal/healthcare/instructions

I hope that the suggested changes help to improve the quality of the article and that they are well received.

Kind regards

Reviewer 2 Report

The article is an interesting study with a good overall report.
I would like to suggest a more extensive and related literature review in the three itens widely evaluated in the study, ie. the definition of pain for use in health concept.

As reffered in the conclusion, it would be good if there is some possibility to describe the roles of each intervenient (line 459).

It can also be improved the way and which data is collected initialy and arround the other evaluation moments.

Some minor language issues could be corrected or better described. 
